# K-Means-Based DNN Algorithm for a High Accuracy VLP System

**Jianli Jin** [1,2]**, Shouwei Wang** [1]**, Lu Yang** [1,2]**, Huimin Lu** [1,2]**, Jianping Wang** [1,*]**, Danyang Chen** [1,2]**, Lifang Feng** [1]**, Hongyao Chen** [1] **and Hongyu Zhang** [1]

[1] School of Computer and Communication Engineering, University of Science and Technology Beijing, Beijing 100083, China; jinjianli@ustb.edu.cn (J.J.); m202220839@xs.ustb.edu.cn (S.W.); g20209478@xs.ustb.edu.cn (L.Y.); hmlu@ustb.edu.cn (H.L.); chendanyang@ustb.edu.cn (D.C.); lffeng@ustb.edu.cn (L.F.); chenhongyao@ustb.edu.cn (H.C.); d202310395@xs.ustb.edu.cn (H.Z.)
[2] Shunde Innovation School, University of Science and Technology Beijing, Foshan 528000, China
* Correspondence: jpwang@ustb.edu.cn

**Abstract:** In this paper, a positioning algorithm based on the combination of K-means clustering and deep neural networks (DNNs) is first presented for multiple light emitting diodes (LEDs) integrated with visible light positioning (VLP) systems. We extracted the maximum value from the collected optical power of LEDs, utilizing the ratio of each optical power to this maximum optical power as the input training data. The experimental results demonstrate that the proposed algorithm outperformed the conventional DNN algorithm in terms of anti-jamming capability and positioning accuracy. In addition, the positioning accuracy of the proposed system reached a millimeter level, which is the highest experimental VLP accuracy, to the best of our knowledge.

**Keywords:** visible light positioning; K-means; deep neural network; light emitting diodes





## 1. Introduction

With the wide deployment of phosphor white light emitting diodes (LEDs) and silicon-based photodetectors (PDs), smart devices using light information have emerged as a potential solution to support Internet of Things (IoT) services. Phosphor white LEDs, created by applying a yellow phosphor coating to blue LEDs to generate white light, offer a substantial benefit through their high quantum efficiency and low cost, crucial for general lighting applications. Therefore, fluorescent LEDs are being considered as optical information emitters for large-scale applications.

Because of the widespread use of phosphor-based LEDs and silicon-based PIN photodetectors, wireless light-based information systems are expected to provide reliable and stable indoor interconnection and positioning services. Over the past decade, wireless positioning systems have become crucial in our daily lives, particularly with the growing demand for indoor positioning services. Accurate positioning is essential for deploying artificial intelligence devices, such as robots, in indoor settings. While global positioning systems (GPS) are prevalent in outdoor environments, their effectiveness is significantly reduced indoors due to signal attenuation through solid walls [1,2]. Various indoor positioning techniques have been developed to address the limitation of GPS, including wireless signal transmission-based methods, wireless local area networks (WLAN) [3], Bluetooth, ZigBee, and infrared-based systems [4,5], as well as radio frequency identification, ultra-wide band (UWB), and other emerging technologies [6,7]. Despite enabling indoor positioning, these techniques face several challenges, including low accuracy, electromagnetic interference, security concerns, and limited spectrum resources, which currently limit their widespread adoption. In contrast, indoor visible light positioning (VLP) techniques have garnered considerable attention due to their multifaceted advantages, such as their cost-effectiveness; enhanced security; and the simultaneous provision of lighting, positioning, and communication [8,9].

Indoor VLP systems utilize LEDs as signal sources to achieve indoor positioning by receiving light signals. The systems are classified into two types based on the receiver: camera-based and PD-based positioning. The camera-based system consists of an array of LEDs and a high frame rate image sensor [10,11], which requires a stabilized camera to reduce jitter. However, this approach, which necessitates constant image capture and processing, imposes substantial requirements on the system memory and commonly encounters difficulties in achieving real-time positioning due to computational intricacy. As a comparison, the PD-based system demonstrates the capacity to rapidly respond to light signals, facilitating a lower latency. Furthermore, it typically obviates the need for intricate image processing hardware and algorithms, thereby streamlining hardware design and diminishing costs.

In VLP systems based on PD positioning techniques, triangulation algorithms are a widely employed positioning scheme that integrates methods such as time of arrival (TOA), time difference of arrival (TDOA), angle of arrival (AOA), and received signal strength (RSS). The study in [12] presents a VLP system employing RSS methodology, enhanced by the importance of the sampling method to reduce the computational complexity. The experimental results highlight this scheme's robustness. The authors of [13] proposed a TOA-based VLP system that can achieve a centimeter-level positioning accuracy. In [14], the AOA positioning algorithm is utilized, enhancing positioning precision at the expense of increased system complexity. The results indicate that this approach achieves a positioning accuracy of up to 10 cm. A TDOA-based VLP system indicates that high sampling rates and precise timing can further complicate the system. The outcomes reveal that this approach attains a positioning accuracy of 3.9 cm [15]. A comparative analysis suggests that the RSS positioning algorithm is more suitable for VLP systems because of its broader applicability. To enhance the positioning accuracy, machine learning has created numerous advancements in the domain of indoor VLP. As detailed in [16], RSS data samples are gathered from various locations during offline training. These samples are then utilized as training data to construct a model that can accurately estimate the position of PD from the new RSS samples received during online positioning, achieving a positioning accuracy of 10.5 cm. A new positioning algorithm based on a long short-term memory fully connected network (LSTM-FCN) has been proposed in [17], for enhanced positioning accuracy in scenarios with multiple LEDs and a single PD in VLP. The approach achieves considerable complexity, albeit with a positioning accuracy of less than 5 cm. In [18], the adaptive residual weighted k-nearest neighbors (ARWKNN) fingerprint positioning algorithm demonstrates an improvement over the traditional K-nearest neighbors (KNN) algorithm in terms of performance. In addition to traditional machine learning algorithms, neural network techniques have also been applied to indoor VLP systems. A multiple-bandwidth generalized regression neural network (GRNN) with the outlier filter indoor positioning approach (GROF) is proposed in [19], which enhances the robustness of the positioning against environmental variations. However, it requires further refinement to improve its accuracy in positioning. A VLP algorithm based on a deep neural network with Bayesian regularization (BR-DNN) is proposed in [20], which achieves positioning with few training points, reducing complexity of the system. However, the accuracy of the positioning needs further improvement. In [21], a dual-layer fusion network-based algorithm is proposed for indoor VLP, which addresses the challenge of signal fluctuations at the receiver due to unstable power output from the light-emitting LEDs. However, the emphasis was not on positioning accuracy, leading to a compromise in positioning precision. In [22], an integrated PD and camera VLP receiver was proposed based on RSS algorithm. In the scheme, two-dimensional and three-dimensional positioning can be achieved using only one PD and one camera under the scenario of three LEDs, but the proposed system can be further improved in terms of positioning accuracy. The aforementioned research successfully implemented the application of machine learning and deep neural networks (DNN) in VLP systems. However, the positioning accuracy could be further improved for most of the VLP systems. Drawing inspiration from insights acquired in previous research,

we propose a novel positioning algorithm based on the combination of K-means clustering and DNN for a VLP system to enhance the positioning accuracy. The main contributions of our work can be summarized as follows:

- For the first time, in this article, in order to achieve high-precision positioning of LED and PD-based indoor devices, we present a novel algorithm based on the combination of K-means clustering and DNNs. This algorithm achieves a better anti-interference capability and higher positioning accuracy compared with conventional DNN algorithms.
- To evaluate and analyze the performance of the proposed system, we developed and executed a comprehensive experimental framework. The VLP system exhibits an excellent performance in terms of interference resistance, and the highest millimeter-level positioning accuracy, to the best of our knowledge.

## 2. VLP System Model

In this section, a comprehensive exploration of the proposed VLP system model is presented. Figure 1 illustrates the model's adaptability to scenarios involving multiple LEDs and a single PD. Initially, signals are transmitted to the LEDs. Subsequently, the PD receives and amplifies the signals. The signals undergo fast Fourier transform (FFT), followed by a two-phase K-means clustering process: the offline phase and the online phase. During the offline phase, a training set and model are established using the received RSS data. In the online phase, the PD's positioning is calculated using the positioning algorithm based on DNN and the trained model, thereby realizing VLP.

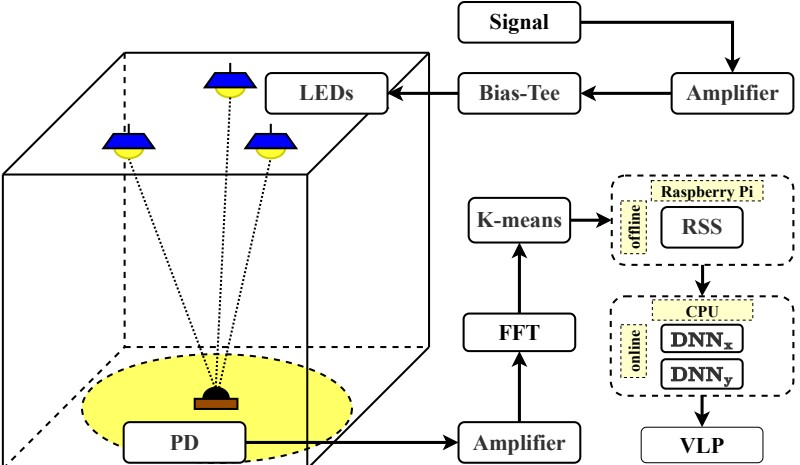

**Figure 1.** The schematic block diagram of the proposed system.

### 2.1. The Channel Model

In the domain of VLP, the influence exerted by directional light is pronounced and plays a pivotal role in system [23]. Given this premise, our investigation is circumscribed to Line-of-Sight (LOS) links exclusively. It is imperative to highlight that, within the conventional indoor illumination spectrum, the inter-symbol interference attributed to multipath is insubstantial and can be ostensibly disregarded. Within the realm of visible light links, the LOS channel gain intertwining the *k*-th LED and PD is articulated by the following equation:

$$G_k = \frac{(m_k + 1)A_r}{2\pi d_k^2} \cos^{m_k}(\phi_k) T_s(\psi_k) g(\psi_k) \cos\psi_k \tag{1}$$

where $m_k = -ln2/ln(cos\phi_{1/2})$ denotes the Lambertian radiation order, $\phi_{1/2}$ represents the half power angle of LEDs, while $A_r$ epitomizes the surface area of PD. The distance between the *k*-th LED and PD is represented by the term $d_k$. The angles $\phi_k$ and $\psi_k$ denote the irradiation and incident angles, respectively. The constructs $T_s(\psi_k)$ and $g(\psi_k)$ represent

the gains of the optical filter and concentrator, respectively. The gain in the channel depends on the unique characteristics of both the LEDs and PD, in conjunction with the prevailing transmission distance, $g(\psi_k)$ is defined as follows:

$$g(\psi_k) = \begin{cases} \dfrac{n^2}{sin^2(\psi_c)} & 0 \leq \psi_k \leq \psi_c \\ 0 & \psi_k \geq \psi_c \end{cases} \tag{2}$$

For PD, the electrical power received from the positioning subcarrier of the *k*-th LED can be expressed as: $P_k^{rec} = G_k P_k$. To elaborate further, assuming that the PD and LED are perpendicular to the ceiling, it follows that $\cos(\phi_k) = \cos(\psi_k) = h/d_k$, where h represents the vertical distance between the LEDs and PD, and thus $G_k$ in (1) can be rewritten as [24]:

$$G_k = \frac{h^{m_k+1}(m_k + 1)A_r}{2\pi d_k^{m_k+3}} T_s(\psi_k)g(\psi_k) = F(m_k + 1)\frac{h^{m_k+1}}{d_k^{m_k+3}} \tag{3}$$

In the equation, where $F(\bullet) = A_r T_s(\psi_k)g(\psi_k)/2\pi$ is a constant depending on the characteristics of the LEDs and PD. The electrical power received by PD can be rephrased as:

$$P_k^{rec} = \frac{P_k F(m_k + 1)h^{m_k+1}}{d_k^{m_k+3}} \tag{4}$$

*2.2. K-Means-DNN Model*

The structure of the proposed K-means DNN is depicted in Figure 2. Firstly, the captured RSS data are clustered by K-means model based on the number of LEDs. In the proposed system, the RSS data are clustered in three groups. After that, the clustered data are sent to DNN containing five hidden layers for positioning. Figure 3 details the flowchart of the K-means DNN algorithm. K-means is an iterative cluster analysis algorithm that involves several steps. Initially, the data are divided into K groups and K samples are randomly selected as the initial cluster centers. The distance between each object and centroid is then calculated, and the objects are assigned to their nearest centroid. The centroids and the objects assigned to them constitute a cluster. For each assigned sample, the centroids are recalculated based on the current members of the cluster. The centroids are recalculated for each assigned sample based on the cluster's current objects. This process persists until specific termination criteria are met, which include minimal or no reassignment of objects to different clusters, minimal or no change in cluster centroids, or local minimization of the sum of squared errors. After performing K-means clustering, the DNN algorithm extracts the optical power of three LEDs in the frequency domain, using it as the input. Subsequent to the computation of the outputs from the hidden layer neurons and the output layer neurons, the loss is ascertained by quantifying the discrepancy between the calculated values and the true values. The loss is then evaluated against a predefined threshold of acceptability. Should the loss not align with the desired benchmark, an update to the parameters of both the hidden and output layers is performed. This iterative process of evaluation and parameter adjustment continues until the conclusion of the training phase. The extraction of LED optical power involves frequency analysis to distinguish frequencies of background or natural light from those of LED transmission signals, thus facilitating the filtering out of background and natural light, which enhances the interference resistance. Subsequently, the data undergo preprocessing, including the normalization of the signals, to mitigate the discrepancies among LEDs.

The positioning algorithm based on the combination of K-means and DNNs has two distinct phases: an online phase and an offline phase. To acquire training data, $m \times n$ coordinate points are selected for placing PD; the received power at each coordinate point is then recorded.

$$P_{ij}^{rec} = \begin{bmatrix} P_{ij1}^{rec} & P_{ij2}^{rec} & P_{ij3}^{rec} \end{bmatrix} \quad 1 \leq i \leq m, 1 \leq j \leq n \tag{5}$$

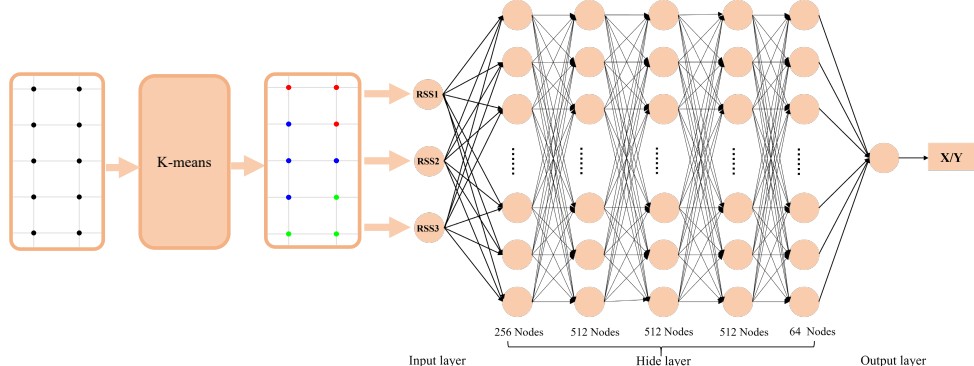

**Figure 2.** The structure diagram of K-means-DNN.

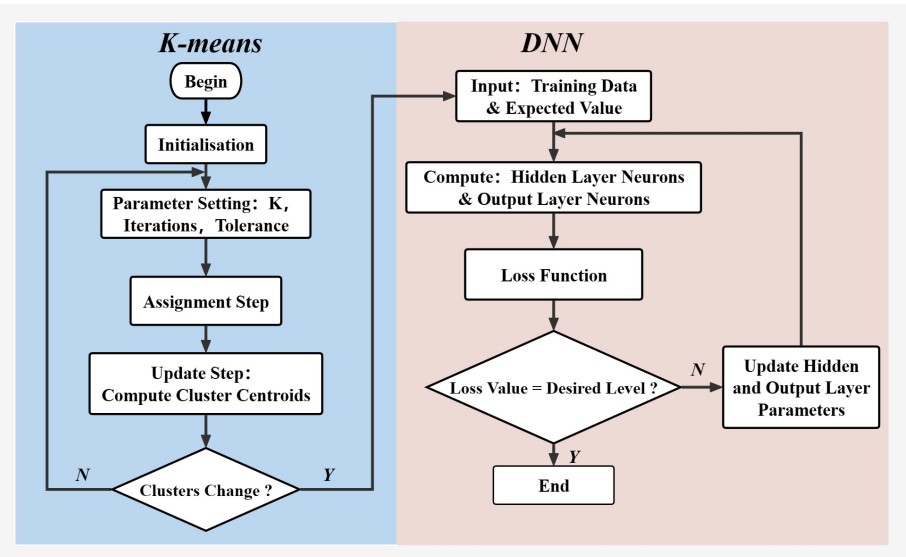

**Figure 3.** The flowchart of K-means-DNN.

where $P_{ijk}^{rec}$ represents the optical power of the *k*-th LED received by PD. Thus, the optical power received by PD from different LEDs can be described as follows:

$$P_{ijk}^{\text{rec}} = \begin{bmatrix} P_{ijk,1}^{\text{rec}} & P_{ijk,2}^{\text{rec}} & \cdots & P_{ijk,t}^{\text{rec}} \end{bmatrix} \tag{6}$$

where *t* denotes the number of consecutively collected optical power data at the same coordinate point. In order to reduce complexity, the experiment independently processes the x and y coordinates of the data points, training them separately. The description of a single training data point for the x-coordinate is as follows:

$$\begin{aligned} \boldsymbol{Train}X_{rk} &= \begin{bmatrix} P_{ij}^{\text{rec}} & x_{rk} \end{bmatrix} \\ &= \begin{bmatrix} P_{ij1,1}^{\text{rec}} & \cdots & P_{ij1,t}^{\text{rec}} & \cdots & P_{ij3,1}^{\text{rec}} & \cdots & P_{ij3,t}^{\text{rec}} & \chi_{rk} \end{bmatrix} \end{aligned} \tag{7}$$

where $x_{rk}$ indicates the x-coordinate of the training data. The Euclidean distance between the estimated point and the training point values can be expressed as:

$$dis_e = \left( \sum_{k=1}^{3} |\hat{x}_{ek} - \hat{x}_{rk}|^2 \right)^{\frac{1}{2}} \tag{8}$$

where $x_{ek}$ represents the fingerprint information of the estimation point of the *k*-th LED. After calculating the Euclidean distances within the feature space of each sample in the

fingerprint database, $dis_e$ is arranged in ascending order. Subsequently, the coordinates of the points corresponding to the first $D$ Euclidean distances $dis_e$ are selected, and the estimation point coordinates $(x_e, y_e)$ are determined by computing the average values of these points.

$$(x_e, y_e) = \frac{1}{D} \sum_{e=1}^{D} (x_e, y_e) \tag{9}$$

The positioning mean square error (*MSE*) of the device is given by the follow:

$$MSE = \frac{1}{D} \sum_{e=1}^{D} [(\hat{x}_e - \hat{x}_r)^2 + (\hat{y}_e - \hat{y}_r)^2] \tag{10}$$

## 3. Experimental Results

Our experimental scenario represents a typical indoor environment with dimensions of 1 m×1 m×2 m containing three LEDs and a PD, as illustrated in Figure 4. To reduce the impact of natural and background light, each LED employs a distinct modulation frequency: 5000 Hz, 6500 Hz, and 7800 Hz, respectively. For a complete description of the remaining parameters, please refer to Table 1.

**Table 1.** The parameters of VLP system.

| Symbol | Parameter | Value |
|--------|-----------|-------|
| $Dis_{led}$ | Adjacent LED distance | 0.85 m |
| $x_r, y_r$ | The coordinates of LEDs | (0.5, 0.5) (0.5, 1.25) (1.5, 1.0) |
| $P_t$ | The emission power of LEDs | 30 W |
| $\Phi$ | Illuminance | 300∼900 Lx |
| $A_{PD}$ | Area of photo diode | 1 cm$^2$ |
| $m$ | Order of Lambertian emission | 0.646 |
| $FOV$ | The field of view of PDs | 70° |

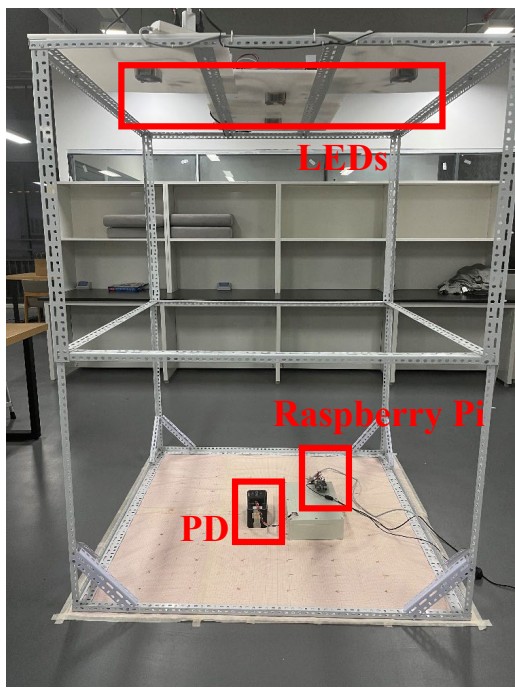

**Figure 4.** The experimental setup of VLP system based on K-means-DNN.

This study presents a comprehensive analysis of the initial phase of the K-means-DNN algorithm, specifically focusing on the K-means clustering component. As a widely applied

machine learning algorithm, K-means plays a key role in positioning systems. Notably, the selection of the number of clusters K is crucial for optimal positioning accuracy.

Figure 5a illustrates that with the DNN performance stable and parameters fixed, when the K values are 2, 3, 4, and 5, the corresponding positioning accuracies are 1.48 cm, 0.78 cm, 0.73 cm, and 0.74 cm, respectively. These outcomes indicate that as the K value increases, the positioning error shows a continuous decreasing trend. Specifically, when the K value increases from 2 to 3, the positioning error significantly reduces from 1.48 cm to 0.78 cm, achieving millimeter-level accuracy. However, when the K value increases to 5, the improvement in the positioning accuracy is limited compared with the increase in computational load and time.

Therefore, after considering both the accuracy improvement and computational efficiency, this paper selected K = 3 as the optimal parameter in the subsequent performance analysis. This decision was based on experimental data, aiming to achieve the best balance between accuracy and computational efficiency in the positioning system. Figure 5b presents the cumulative distribution function (CDF) curves of the positioning errors along the x-axis and y-axis. The analysis reveals that the CDF curve of the x-axis exhibited a higher degree of smoothness. While the CDF curve for the y-axis showed slight fluctuations in certain regions, it generally tended towards smoothness. By comparing and analyzing errors along the x-axis and y-axis, it can be observed that the average positioning error of the x-axis was lower than that of the y-axis. This can be attributed to the uneven indoor lighting environment where the interference area from lighting fluorescent tubes were rectangles even if the positioning LED was square.

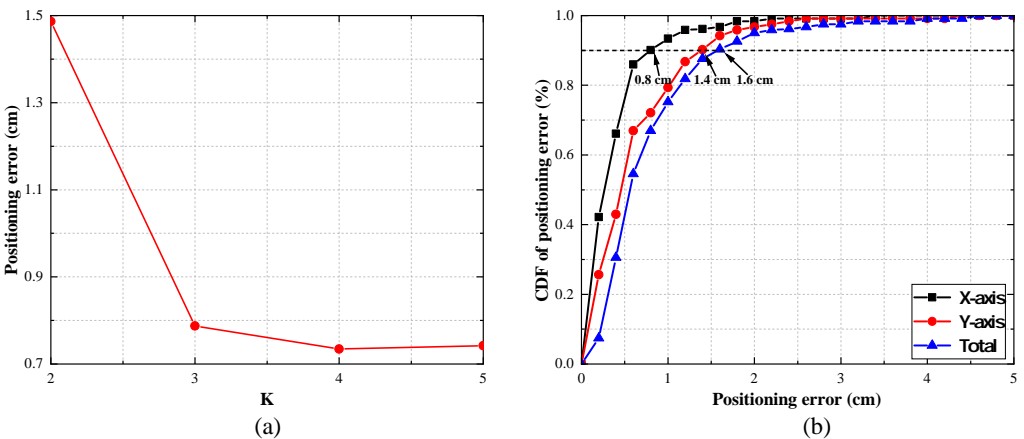

**Figure 5.** The proposed positioning error with (**a**) different value of K (**b**) CDF of positioning errors.

We compared the performance between the K-means-DNN algorithm and the conventional DNN algorithm in the same indoor VLP system. The system adopted multi-LEDs and single PD environment and both algorithms used the same RSS training dataset. The results showed that the average positioning errors of the K-means-DNN algorithm and the conventional DNN algorithm were 0.78 cm and 1.67 cm, respectively. It can be observed that K-means-DNN outperformed the conventional DNN in positioning error. Figure 6 demonstrates the CDF of the VLP system based on K-means-DNN and conventional DNN. It can be observed that the positioning errors at a 90% confidence interval were 1.6 cm and 3.2 cm for the respective methods. For the proposed K-means-DNN algorithm, approximately 75% of the errors were less than 1 cm and 95% of the positioning errors were less than 2 cm, with only a minimal number exceeding this margin. In contrast, in the DNN algorithm, 33% of the positioning errors were below 1 cm and the worst error exceeded 8 cm.

Figure 7a,b illustrate the performance comparison of two algorithms in the CDF charts for both the x and y-axis, respectively. In the context of positioning accuracy, the K-means-DNN algorithm outperformed the DNN algorithm, achieving approximately 95% accuracy in positioning errors under 1 cm on the x-axis, compared with about 85% for the DNN algorithm. Furthermore, the K-means-DNN algorithm maintained approximately

95% for positioning errors under 2 cm on the y-axis. In contrast, the DNN algorithm showed a marked decrease in accuracy, achieving only 60%. The K-means-DNN algorithm showed a significantly better positioning performance than the DNN algorithm in both the x and y-axis.

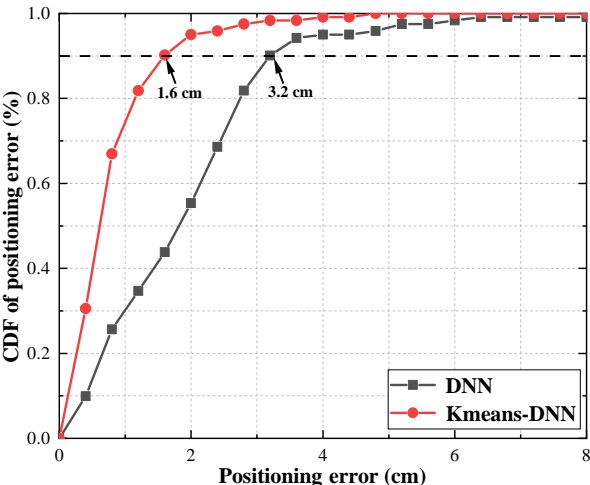

**Figure 6.** The CDFs of different positioning methods.

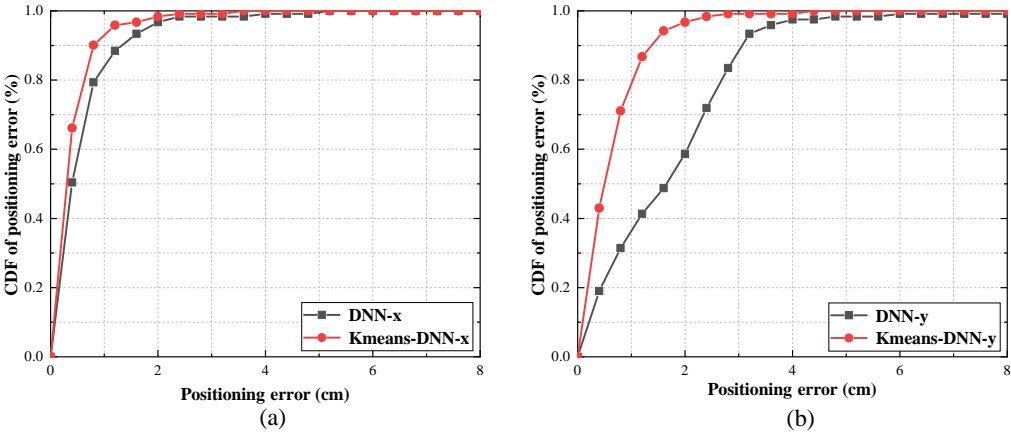

**Figure 7.** The CDFs of different positioning methods for (**a**) x-axis and (**b**) y-axis.

Figure 8 displays the positioning coordinate diagram of the K-means-DNN and conventional DNN algorithms. The blue square points indicate the actual positioning coordinates, while the red circular points signify the estimated coordinates of the algorithms. The comparison reveals a greater overlap between blue and red points in the K-means-DNN algorithm than in the conventional DNN algorithm. As shown in Figure 8a, in the K-means-DNN algorithm, the errors are mostly focused on individual points located at the edges. Conversely, as shown in Figure 8b, the conventional DNN algorithm tends to produce larger errors on most positioning points randomly, even if some positioning results are relatively precise. Thus, regarding individual positioning point outcomes, the K-means-DNN algorithm surpasses the DNN algorithm.

To provide a visual comparison of the performance of the K-means-DNN and conventional DNN algorithms, we present error bar charts for both algorithms. Figure 9 shows the error distribution at 121 positioning points, with Figure 9 representing the K-means DNN and conventional DNN algorithms, respectively. In the bar chart, the depth of color and height indicate the range of errors at each point. As shown in Figure 9a, the maximum error for the K-means-DNN algorithm is 4.44 cm. For the conventional DNN algorithm shown in Figure 9b, the maximum error is 8.32 cm. This indicates that in a VLP system, using the K-means-DNN algorithm as opposed to the conventional DNN algorithm alone

provides a more accurate positioning performance. The output results show smaller errors and better stability for the positioning system, approaching the true coordinates.

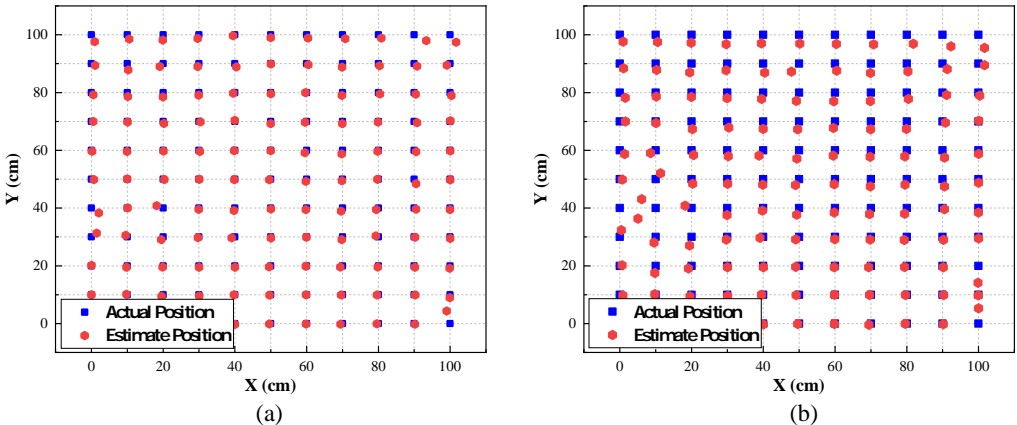

**Figure 8.** The VLP positioning coordinates of (**a**) K-means-DNN and (**b**) conventional DNN.

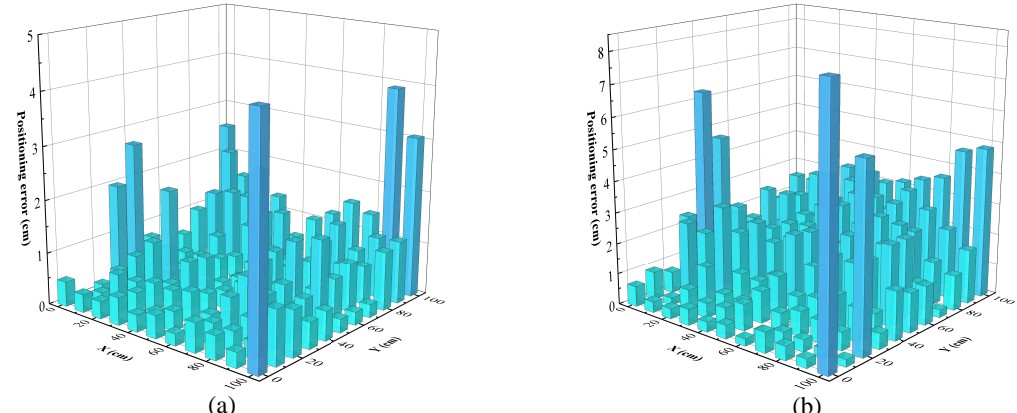

**Figure 9.** The positioning error histogram of (**a**) K-means-DNN and (**b**) conventional DNN.

Please consult Table 2 for a detailed comparison of the work.

**Table 2.** Comparative Analysis of Positioning Accuracy and Complexity Across Various Studies.

| References | Methodology | Complexity | Positioning Accuracy |
|---|---|---|---|
| This Work | K-means-DNN | Medium | 0.78 cm |
| [12] | Luminance Distribution Model | Medium | 7 cm |
| [13] | Cramer-Rao Bound | Medium | 7 cm |
| [14] | Maximum Likelihood | Medium | 10 cm |
| [15] | Pilot Signals | Low | 3.9 cm |
| [16] | Cayley–Menger | Medium | 10.5 cm |
| [17] | LSTM-FCN | Low | 0.92 cm |
| [18] | ARWKNN | Medium | 3.8 cm |
| [19] | GRNN | Low | 0.96 cm |
| [20] | BR-DNN | Medium | 4.5 cm |

## 4. Conclusions

This article investigates the positioning accuracy challenges in VLP systems. Conventional DNN algorithms limit the positioning accuracy of VLP systems, necessitating the development of a novel, more effective algorithm. We introduce an advanced VLP system employing the K-means-DNN algorithm, which incorporates a training set that gathers received power data from 121 distinct points. Additionally, to reduce ambient noise interference and improve the VLP system's resilience, we applied varying frequencies to the

three LEDs. The experimental results demonstrate the system's exceptional performance, highlighting its significant improvements in positioning accuracy. The system achieves an average positioning error of 0.78 cm and a maximum error of less than 4.5 cm, evidencing its precision in positioning accuracy.

**Author Contributions:** Conceptualization, J.J.; Math analysis, S.W.; Investigation, L.Y.; Mathematical Modeling, H.L. and D.C.; Writing, J.W.; Writing—review and editing, L.F.; Data analysis, H.C.; Software, H.Z. All of the authors reviewed the results and approved the final version of the manuscript.

**Funding:** This work was supported in part by the Guangdong Basic and Applied Basic Research Foundation, China (2022A1515110154, 2021B1515120086, 2022A1515110770), Fundamental Research Funds for the Central Universities, China (FRF-TP-22-049A1, FRF-TP-22-044A1), National Natural Science Foundation of China (62374015).

**Institutional Review Board Statement:** Not applicable.

**Informed Consent Statement:** Not applicable.

**Data Availability Statement:** The data presented in this study are available upon request from the corresponding author.

**Conflicts of Interest:** The authors declare no conflict of interest.

## Abbreviations

The following abbreviations are used in this manuscript:

| | |
|---|---|
| LED | Light Emitting Diode |
| DNN | Deep Neural Network |
| PD | Photo Detectors |
| VLP | Visible Light Positioning |
| IoT | Internet of Things |
| RGB | Red-Green-Blue |
| GPS | Global Positioning System |
| WLAN | Wireless Local Area Networks |
| UWB | Ultra-Wide Band |
| TOA | Time of Arrival |
| AOA | Angle of Arrival |
| TDOA | Time Difference of Arrival |
| LSTM-FCN | Long Short-Term Memory Fully Connected Network |
| ARWKNN | Adaptive Residual Weighted K-nearest Neighbors |
| GRNN | Generalized Regression Neural Network |
| BR-DNN | Deep Neural Network with Bayesian Regularization |
| FFT | Fast Fourier Transform |
| LOS | Line-of-Sight |
| CDF | Cumulative Distribution Function |

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
