# Peer review of "K-Means-Based DNN Algorithm for a High Accuracy VLP System"

_photonics, doi:10.3390/photonics11030209_

Round 1

Reviewer 1 Report

Comments and Suggestions for Authors

(1)     The figures can be refined to make them clear. Such as figure 9.

(2)     The authors need to check all the equations carefully. The functions should not be italic.

(3)     Using a table to compare the work with relative works is necessary.

(4)     The authors should denote the results clearly whether it is simulation or experiment.

(5)     More relative works should be cited in the introduction. Such as:

 “3D NLOS VLP Based on a Luminance Distribution Model for Image Sensor”, IEEE Internet of Things Journal, vol. 25, no. 8, pp. 6902-6914.

 “Experimental Demonstration of an Indoor VLC Positioning System Based on OFDMA”, IEEE Photonics Journal, vol. 9, no.2, pp. 7902209, 2017.

Comments on the Quality of English Language

English need to be improved.

Reviewer 2 Report

Comments and Suggestions for Authors

The paper proposes a positioning algorithm that combines K-means clustering and deep neural networks (DNNs) for multiple light-emitting diodes (LEDs) visible light positioning (VLP) systems. Consideration for publication is suggested after implementing the following changes:

1. It is recommended to omit the word "using" from the paper title.

In the literature review section of the introduction on page two, please provide the accuracy achieved by the surveyed studies to emphasize the contribution of the study.

2. Please clarify the nature of the study, as Figure 4 depicts an experimental setup. In addition, the abstract and conclusion refer to results as experimental and simulation, while the discussion in results section only presented simulation data. If experimental results are not available, the authors are encouraged to conduct experiments to validate the proposed method.

3. Please ensure the caption for Figure 4 accurately reflects the content depicted.

4. On page 7, lines 202 to 205, where the authors attribute the error asymmetry to uneven indoor lighting, please consider whether reflection from walls and ceilings, which was disregarded in the manuscript, may also contribute.

Comments on the Quality of English Language

It is recommended to review the manuscript for any grammar and language errors. For instance, it is suggested to combine the first two sentences of the original contribution's first bullet point into a single sentence.

Reviewer 3 Report

Comments and Suggestions for Authors

This paper proposes a k-means based DNN algorithm using for VLP system. Experimental results show that the proposed algorithm could achieve the highest positioning accuracy among the existing VLP systems, This work is both novel and interesting but some questions should be addressed before publication.

1, How does this approach compare in complexity and computational efficiency with existing algorithms?

2, Authors have referenced relevant works, but a more detailed comparison with existing methods could strengthen the paper.

3, The results demonstrate improvement over conventional DNN algorithm. More details about the improvement should be discussed.

4, LED abbreviation in line 36 is repeated.

5, Is figure2 incorrectly labelled?

Comments on the Quality of English Language

N/A
